# PatchSAGE: A Probe-Based Detector Using Saliency Alignment, Gradients, and Layer Sensitivity

## Abstract

Adversarial patches cause targeted misclassification by steering a model's evidence toward a small, visible region while human perception remains largely unaffected. We propose PatchSAGE, a post hoc, model-agnostic detector that attaches lightweight probes to a frozen classifier and fuses three complementary signals: (i) input-gradient statistics of the predicted class, (ii) layer-wise sensitivity to small activation noise, and (iii) human–model saliency alignment, quantified by comparing Grad-CAM with human saliency maps. Features from these probes are fed to a small secondary classifier (detector) that predicts whether an input is patched. To our knowledge, PatchSAGE is the first adversarial-patch detector to explicitly incorporate human attention modeling via saliency alignment, aligning what the model relies on with where humans look, without modifying or retraining the base model. Across CAT2000, FIGRIM, and SALICON, using ResNet-50 and EfficientNet-B0 backbones, PatchSAGE achieves F1 scores up to 99.6% and remains in the 90–99% range across settings, outperforming probing baselines, SentiNet, and X-Detect in our setting. Ablations show monotonic gains from adding gradients and alignment to sensitivity, indicating complementary cues and highlighting alignment's discriminative power. PatchSAGE is simple to deploy (post hoc; no retraining) and provides interpretable rationales via its saliency and sensitivity components, suggesting a practical path to robust, explainable detection of adversarial patches.

## 1 Introduction

Adversarial patch attacks involve embedding a localized, often conspicuous perturbation (e.g., a sticker) into an image to mislead a deep neural network's prediction while leaving human perception largely unchanged Brown et al. (2017). Defending against such localized attacks is challenging because the adversary's modification is not a small, dispersed noise (as in classic $\ell_p$ attacks) but a concentrated region that can drastically alter model features. Crucially, these attacks exploit discrepancies between model and human vision; the model's decision can hinge on the adversarial patch, whereas humans ignore it and classify the image based on the true object.

A straightforward approach is to detect adversarial inputs rather than attempting to correct them. However, adversarial example detection has proven notoriously difficult. Prior studies have shown that many proposed detectors can be defeated by adaptive attacks that specifically target their detection cues Carlini & Wagner (2017). In fact, characteristics once thought intrinsic to adversarial examples (e.g., abnormal activation statistics or input deviations) can often be masked by a sufficiently sophisticated adversary Tramer et al. (2020). This arms race underscores the need for detection methods that capture fundamental differences in model behavior when an adversarial patch is present, yet are general enough to handle novel attacks.

Our key insight is that adversarial patches induce inconsistent model behavior that can be revealed through a fusion of interpretability, saliency alignment, and sensitivity analysis. We hypothesize that by probing the model's internal responses – measuring how input gradients and intermediate feature sensitivities change – and by comparing model attention with human attention, we can capture revealing signs of patch attacks.

In this paper, we introduce a probe-based adversarial patch detector that systematically combines these complementary signals. Small probe modules attached to the base classifier extract a rich set of features from a given input: (1) Gradient statistics such as the mean and maximum absolute gradient of the predicted class score with respect to the input pixels, which tend to be anomalously large or localized under patch attacks; (2) Activation sensitivity metrics that quantify how sensitive the final prediction is to perturbations in each layer or region (e.g., how much the output changes if we inject noise or mask activations at a given layer), capturing the model's brittleness when a patch dominates the features; and (3) saliency-based interpretability divergence between the model's visual explanation and human vision, measured by comparing the model's Grad-CAM heatmap with a ground-truth human saliency map for the image. Intuitively, a clean image should have a reasonable overlap between model and human salient regions, whereas an image with an adversarial patch will show the model fixating on the patch (high model saliency there) which humans would not, yielding a large divergence.

While prior adversarial patch detection methods such as SentiNet, Metzen's auxiliary detectors, and X-Detect leverage either saliency or internal representations independently, they often suffer from limited interpretability or susceptibility to adaptive attacks. Our approach departs from these by explicitly integrating human-aligned interpretability, i.e., saliency alignment, into the detection pipeline. We fuse three complementary modalities: layer-wise sensitivity, gradient statistics, and human-vs-Grad-CAM saliency divergence, to create a detection mechanism that not only distinguishes adversarially patched images with high accuracy but also offers transparency into why a prediction was flagged. To our knowledge, this is the first method to systematically combine human saliency alignment with gradient and activation-based cues, demonstrating that interpretability and robustness are not in tension, but mutually reinforcing. This enables us to set a new state-of-the-art in both detection performance and qualitative explainability.

## 2 RELATED WORK

Adversarial Example Detection: A variety of detectors have been proposed for adversarial examples, though many focus on $\ell_p$ perturbations rather than patch attacks. Classical detection methods include augmenting the classifier with an additional outlier/adversary output class Grosse et al. (2017), anomaly-modeling detectors (e.g., GAN-based) Wang et al. (2023), uncertainty-based Bayesian detectors Li et al. (2024) and feature-statistics detectors **?**, training a separate binary classifier on inputs (or on transformed inputs) to flag adversaries, and inserting small sub-networks into the victim model to monitor its internal activations. For instance, Metzen et al. train a small CNN on features from an intermediate layer of a vision model to detect adversarial perturbations Metzen et al. (2017). Bhagoji et al. Bhagoji et al. (2017) reduce the input dimension via PCA and train a fully-connected network on the compressed input as a detector, and Li and Li Li & Li (2017) model convolutional filter statistics to distinguish adversarial from benign inputs.

Statistical and Distributional Anomalies: Some works detect adversarial examples by identifying distributional shifts. Grosse et al. Grosse et al. (2017) use statistical tests (e.g., maximum mean discrepancy) and outlier classes to flag inputs off the data manifold. Feinman et al. Feinman et al. (2017) apply kernel density and Bayesian uncertainty estimates in the feature space to detect anomalies. These model-agnostic methods are lightweight but can be bypassed by adaptive attacks. Still, the core idea—that adversarial inputs differ statistically—motivates our use of gradient and activation statistics to capture measurable deviations in model behavior.

Interpretation-Based Defenses: Recent studies have explored the intricate connection between interpretability and adversarial robustness. **?** argue that deep networks do not learn robust, semantically meaningful representations, as adversarial examples can easily disrupt apparent interpretability. They propose adversarial training with a consistency loss to align neuron responses more closely with human-understandable concepts, allowing users to trace errors back to specific neurons and gain insight into model decisions. Prior work has also examined the intricate relationship between adversarial robustness and saliency map interpretability, highlighting how robust models tend to produce more aligned and faithful saliency maps Mangla et al. (2020). Orthogonally, architectural choices can also influence human–model alignment of explanations: for example, dilated convolutions with learnable spacings have been shown to yield Grad-CAM maps that more closely match human saliency (Chamas et al., 2024).

Liu et al. (2021) use explanations to guide adversarial sample generation for robust retraining, highlighting that understanding model decisions reveals vulnerabilities. SentiNet Chou et al. (2020) directly uses Grad-CAM to detect localized universal attacks by identifying salient regions, transplanting them onto benign images, and checking if they induce misclassifications—hallmarks of adversarial patches. SentiNet requires no additional training or prior attack knowledge, using the model's own generalizable attention as a signal of attack. Inspired by this, our method instead quantifies the misalignment between model and human attention, i.e., saliency alignment, as a numeric feature, offering a continuous measure of anomaly in focus and addressing the interpretability gap left by prior works.

Similarly, the paper Boopathy et al. (2020) demonstrates that enhancing interpretability, particularly through guided backpropagation and integrated gradients, improves robustness against adversarial attacks. Their results suggest a synergistic benefit where interpretability serves both as a defense signal and a training regularizer.

On the other hand, several works such as Heo et al. (2019) caution that interpretability tools can themselves be manipulated. These attacks on interpretation reveal that even saliency methods can be co-opted by adversaries to present misleading explanations, suggesting a limit to how much trust interpretability alone can guarantee.

Our work shares the motivation of leveraging model explanations, but instead of relying on post hoc visualizations or interpretability-enhanced training, we define measurable proxies—such as saliency-human alignment and activation shifts—as compact, input-level features for adversarial detection.

Probe Networks and Layer Behavior: Another defense strategy is to monitor internal activations for anomalies. Metzen et al. Metzen et al. (2017) proposed training a subnetwork on features from a specific layer, which Rounds et al. Rounds et al. (2020) extend in probing by attaching probe CNNs to every layer. Each probe compresses activations into a fixed-size output, and all are concatenated and classified as "adversarial" or "benign." This approach, trained across attacks, captures generalizable activation patterns and achieves higher detection rates than single-layer methods. Our method also follows a probing philosophy but uses analytical probes—such as gradients, sensitivity, and saliency misalignment—rather than trainable CNNs. This yields compact, interpretable features without modifying or retraining the model, making our approach post hoc and model-agnostic. While Rounds et al.'s approach may better capture subtle distribution shifts, our probes target high-level, intuitive discrepancies with minimal overhead.

Gradient-based Works: Adversarial patches similar to adversarial perturbations are identified as features Ilyas et al. (2019) that produce extremely high gradient magnitudes localized to the patch—unlike benign images with smoother, broader gradient distributions. Prior work such as Gradient Similarity detectors Dhaliwal & Shintre (2018) and SentiNet Chou et al. (2020) confirm that leveraging gradient summary statistics effectively distinguishes patched inputs.

Adversarial Patch Defenses: For adversarial patch attacks, certified defenses like Minority Reports McCoyd et al. (2020) adopt a proactive masking strategy. By occluding different regions of an image and analyzing prediction consistency, the method identifies clusters of correct predictions (true label) that emerge when the patch is masked—signaling an attack. This ensemble approach provides formal guarantees for patches below a certain size, offering strong robustness with low false positives and modest clean accuracy loss. While our method does not offer certification, it similarly avoids assuming patch location, instead detecting indirect signals of tampering through gradient, activation, and saliency-based probes.

X-Detect Hofman et al. (2024) targets physical patch attacks in object detection by combining two patch-resilient base detectors: one uses segmentation and a prototype-based classifier, the other perturbs input images to disrupt patch effects. Disagreement between these detectors and the original model flags a potential attack. With reported near-zero false positives and interpretable outputs, X-Detect emphasizes real-world deployment and explainability. Unlike our lightweight, classifier-focused approach that aligns model and human saliency, X-Detect requires multiple models and is tailored to object detection. Still, both methods reflect a growing emphasis on detection systems that not only identify attacks but also explain their reasoning.

## 3 METHODS

Our detection framework augments a given pretrained classifier with probes that extract diagnostic features from the model's response to an input. These features are then input to a separate lightweight classifier (the detector) that predicts whether the image is clean or contains an adversarial patch. The overall pipeline is illustrated as follows (in a conceptual sense):

**Base Classifier (Frozen Model $f$):** This is the original vision model which we suspect could be attacked. We do not modify $f$'s weights. Given an input image $x$, the model produces a predicted class $\hat{y} = f(x)$ along with intermediate feature maps at each layer.

**Feature Extraction Probes:** We extract three categories of features from $x$ with respect to $f$:

**Gradient-Based Statistics:** We compute the gradient of the model's output score for the predicted class $\hat{y}$ with respect to the input image pixels, $\nabla_x f_{\hat{y}}(x)$. For efficiency, one can use the final linear layer's gradient backpropagated to the input. From this gradient map, we derive summary statistics. In our implementation, we take the mean, maximum, minimum, and standard deviation of the gradients' vector. Intuitively, a normal image typically yields moderate gradient magnitudes distributed across the object, whereas an adversarial patch can create extremely large gradients concentrated in the patched region Dhaliwal & Shintre (2018). For example, if a sticker is driving the classification, changing pixels on that sticker will drastically affect the output, whereas clean images might have lower, more spread-out gradients. These scalar features capture the intensity of the model's reliance on specific pixels. We also record the spatial location of the maximum gradient as a percentage of image area (to indicate whether the strongest gradient is concentrated in a small area), although this also relates to the saliency divergence feature described later.

**Activation Sensitivity:** We probe the model's layer-wise robustness by introducing small perturbations at various layers and measuring the impact on the output using a held-out subset of the dataset. Specifically, for each major convolutional block or layer $L_i$ of the network, we perform a controlled perturbation and see how much the predicted class confidence changes. A simple method is to add a small Gaussian noise $\epsilon$ to the activation map at $L_i$ and run the perturbed input forward to get a new prediction $\hat{y}i'$. We then compute the drop in the original class's logits.

Adversarial patches tend to make the network's decision-making fragile in certain layers. While patch attacks are structured and localized rather than stochastic, perturbation sensitivity remains a valid proxy for fragility. In adversarial learning literature, adversarial noise propagation research demonstrates that injecting noise into hidden layers reveals which layers disproportionately amplify adversarial signal Liu et al. (2021). Likewise, Parametric Noise Injection (PNI) methods show that Gaussian noise at intermediate layers can expose and regularize the internal features most vulnerable to adversarial perturbations, including structured ones like patches He et al. (2019).

For instance, if a patch strongly activates some filters, disrupting those filters slightly might cause a large output change, indicating an anomalously high sensitivity. We record the sensitivity values for a selection of layers (each ResNet block). These values serve as features that characterize the stability profile of the network for the given input. We expect clean images to have a more uniform or lower sensitivity, while patched images might show a spike at the layers where the patch's features dominate.

**Grad-CAM–Human Saliency Divergence:** This feature captures the misalignment between model focus and human intuitive focus. We first generate a Grad-CAM Selvaraju et al. (2017) heatmap for the predicted class $\hat{y}$, which highlights regions in the image that most strongly influence the model's prediction. Grad-CAM works by using the gradients of $\hat{y}$ with respect to intermediate feature maps to weight those feature maps, producing an upsampled heatmap that localizes important image regions for the decision. Separately, for the same image $x$, we obtain a human saliency map. In our case, since we use images from datasets with eye-tracking ground truth (CAT2000 Borji & Itti (2015), FIGRIM Bylinskii et al. (2015), SALICON Jiang et al. (2015)), we have a density map of where human viewers tend to look in the scene (for a general task of free viewing). If ground-truth human saliency is not available, one could use a state-of-the-art computational saliency model to predict a human-like attention map for the image. We then compare the Grad-CAM map $G(x)$ with the human saliency map $H(x)$. There are several ways to measure the divergence: we can compute the Pearson correlation between $G$ and $H$ (low correlation means the model is looking at different places than a human expects), or compute a Kullback–Leibler (KL) divergence Kullback

(1951) if we treat them as probability distributions over image pixels. We choose Intersection over Union (IoU) Nowozin (2014) after thresholding both maps; IoU ranges from 0 (no overlap) to 1 (perfect overlap). For a clean image without a patch, we anticipate that the model, if it's functioning normally, will focus on the actual object or salient region that a human might also find relevant (not always true, but often models do pick up on prominent objects). Thus $G$ and $H$ should have a decent overlap, yielding a relatively high correlation. In contrast, an adversarial patch is usually an artificial pattern not meaningful to humans; it might even be intentionally placed in a location that a person wouldn't normally focus on (to avoid detection). The model, however, locks onto it to make its decision. This yields a low correlation (or high divergence) between $G(x)$ and $H(x)$. For example, in an image of a dog with a sticker on the corner, the Grad-CAM for a fooled model might highlight the sticker area strongly, whereas human saliency is centered on the dog's face – the divergence feature will be large. We use 1 minus the chosen alignment score (e.g., $1 - \text{CC}$ or $1 - \text{IoU}$) as a divergence feature, and optionally the fraction of Grad-CAM mass that falls on low-human-saliency regions as a secondary measure.

### 3.1 DETECTION CLASSIFIER TRAINING:

We gather a dataset of such feature vectors by applying the above probes to a mix of benign images and adversarial patch images. In our experiments, we leverage the CAT2000, FIGRIM, and SALICON datasets' images and saliency maps. We synthetically apply adversarial patches to a subset of these images. We then compute $\phi(x)$ for each clean and patched image. Using this data, we train a binary classifier $D(\phi(x))$ that outputs attack present vs clean. We explored both a simple logistic regression (for greater interpretability) and a small multi-layer perceptron as $D$. The classifier training is standard supervised learning, minimizing binary cross-entropy. We take care to prevent overfitting by using cross-validation, given the relatively low-dimensional feature space and the fact that our positive samples might have some biases (since they are generated patches). The outcome is a learned detector that can take a new image, extract the probe features, and predict if a patch attack is likely.

### 3.2 DETECTION INFERENCE:

At test time, for any new input image, we run the same feature extraction process and then feed the feature vector to $D$. The speed of this process depends on how many perturbations and forward/backward passes are needed. In our implementation, we used one backward pass for the input gradient, one forward pass for Grad-CAM computation. Thus, while not as fast as a single forward pass defense, our method could be deployed in an offline filtering scenario or on powerful machines for real-time analysis (especially if we reduce the number of probes or optimize the implementations).

---

**Algorithm 1** Adversarial Patch Detection using Weighted Activations, Gradients, & Saliency Alignment

---

**Require:** pretrained classifier $f$, human saliency map generator $h$, set of test images $\mathcal{X}$, patch detection classifier $g$

1: **for** $x \in \mathcal{X}$ **do**
2:     $y \leftarrow \arg\max f(x)$
3:     Compute Grad-CAM saliency: $s_{\text{model}} = \text{Grad} - \text{CAM}(f, x, y)$
4:     Load human saliency: $s_{\text{human}} = h(x)$
5:     Compute alignment score: $a = \text{IoU}(s_{\text{model}}, s_{\text{human}})$
6:     For each layer $\ell$:
7:         Perturb activations slightly and compute confidence drop: $\Delta_\ell$
8:         Weighted activation feature: $w_\ell = \Delta_\ell \cdot \mathbb{E}[\|f_\ell(x)\|]$
9:     Compute input-gradient norm: $g = \|\nabla_x f(x)_y\|$
10:     Form feature vector: $\mathbf{z} = [w_1, \ldots, w_L, \ g, \ a]$
11:     Predict detection label: $\hat{d} \leftarrow g(\mathbf{z})$
12:     Output: clean if $\hat{d} = 0$, adversarial if $\hat{d} = 1$
13: **end for**

---

Table 1: Detection performance comparison across methods on CAT2000, FIGRIM and SALICON datasets. Metrics are Precision, Recall, and F1 score. Bold values indicate best performance in each row. **Abbreviations:** WA = Weighted Activations; WA+Grad = WA + Gradient Statistics; WA+Grad+Align = WA + Grad + Saliency Alignment.

| Dataset | Model | Metric | Baselines | | | Our Method | | |
|---------|-------|--------|---------|----------|----------|-----|------------|------------------|
| | | | Probing | SentiNet | X-Detect | WA | WA+ Grad | WA+ Grad+ Align |
| CAT2000 | ResNet-50 | Precision | 93.76 | 49.84 | 51.08 | 82.00 | 82.31 | **99.48** |
| | | Recall | 93.63 | 97.80 | 50.12 | 83.86 | 81.01 | **99.81** |
| | | F1 | 93.62 | 66.40 | 51.00 | 83.10 | 84.01 | **99.64** |
| | EfficientNet-B0 | Precision | 88.18 | 51.14 | 74.98 | 81.98 | 91.01 | **91.40** |
| | | Recall | 87.62 | 97.05 | 51.20 | 81.64 | **90.60** | 88.17 |
| | | F1 | 87.58 | 66.21 | 34.00 | 81.58 | **90.90** | 89.47 |
| FIGRIM | ResNet-50 | Precision | 47.59 | 48.88 | 49.00 | 81.43 | 81.17 | **81.74** |
| | | Recall | 49.02 | 68.50 | 50.04 | 81.46 | **95.63** | 86.62 |
| | | F1 | 38.98 | 57.05 | 36.66 | 80.31 | **89.79** | 82.30 |
| | EfficientNet-B0 | Precision | 23.00 | 50.42 | 52.34 | 75.56 | 74.90 | **95.09** |
| | | Recall | 22.37 | 47.24 | 49.99 | 83.93 | 92.37 | **94.86** |
| | | F1 | 23.01 | 48.78 | 34.30 | 77.69 | 81.58 | **94.23** |
| SALICON | ResNet-50 | Precision | 70.01 | 64.58 | 26.00 | 93.66 | **94.18** | 91.74 |
| | | Recall | 63.36 | 74.50 | 51.89 | **98.52** | 96.29 | 94.26 |
| | | F1 | 59.00 | 69.18 | 35.06 | **96.01** | 95.15 | 92.83 |
| | EfficientNet-B0 | Precision | 95.00 | 91.50 | 63.02 | 93.82 | 92.91 | **93.92** |
| | | Recall | 96.72 | 37.87 | 54.90 | **97.26** | 94.60 | 95.53 |
| | | F1 | 97.01 | 53.54 | 43.56 | 92.44 | 93.67 | **93.85** |

To generate adversarial patches, we adapt the Expectation over Transformation (EoT) framework to craft robust, transferable, and universal triggers. Each patch is a square region occupying approximately 2% of the image area, consistent with the scale of localized, realistic threats. Patches are placed at random spatial locations within the image bounds, subject to boundary checks to prevent clipping. The patch is optimized against the frozen image classifier using a cross-entropy loss to maximize confidence in a randomly selected fixed target class. During inference, the finalized patch is overlaid on held-out test images and the resulting patched images are passed through the detector for evaluation.

## 4 EXPERIMENTS

### 4.1 QUANTITATIVE EVALUATION

We evaluated the proposed detector on CAT2000, FIGRIM, and SALICON, using human saliency maps as ground truth for our Grad-CAM alignment measure. Adversarial patches were generated against pretrained ImageNet classifiers (ResNet-50 and EfficientNet-B0) using an expectation-over-transformations variant, and the detector was tasked with separating patched from clean images. We report precision, recall, and F1.

With the full feature set (WA+Grad+Align), the detector attains 99.64% F1 on CAT2000/ResNet-50, 89.47% F1 on CAT2000/EfficientNet-B0 (with best precision 91.40% though F1 is slightly surpassed by WA+Grad at 90.90%), 82.30% F1 on FIGRIM/ResNet-50 (again WA+Grad leads at 89.79%), and 94.23% F1 on FIGRIM/EfficientNet-B0 (best). On SALICON, WA yields the strongest ResNet-50 F1 (96.01%), with WA+Grad at 95.15% and WA+Grad+Align at 92.83%; for EfficientNet-B0, WA+Grad+Align leads at 93.85% (vs. 93.67% for WA+Grad and 92.44% for WA).

These trends reflect dataset/backbone interactions: the alignment term often boosts precision and stabilizes decisions, while in some settings a two-cue variant (WA+Grad) attains the highest F1.

On CAT2000/ResNet-50, baselines are markedly lower (Probing 93.62% F1; SentiNet 66.40%; X-Detect 51.00%), whereas WA+Grad+Align reaches 99.64%. On FIGRIM/EfficientNet-B0, our full model achieves 94.23% F1, dominating Probing (23.01%), SentiNet (48.78%), and X-Detect (34.30%). For SALICON/EfficientNet-B0, the Probing baseline is very strong (97.01% F1), and our method remains competitive (93.85%), while providing the added benefit of human-model alignment diagnostics.

The rows for WA and WA+Grad clarify contributions. On CAT2000/ResNet-50, F1 climbs from 83.10% (WA) → 84.01% (WA+Grad) → 99.64% (WA+Grad+Align). On CAT2000/EfficientNet-B0, WA+Grad edges out the full model (90.90% vs. 89.47%), whereas on FIGRIM/EfficientNet-B0 and SALICON/EfficientNet-B0, adding alignment yields the best F1 (94.23% and 93.85%). Overall, the cues are complementary: gradients and sensitivities expose brittle, patch-driven computation, while alignment verifies that the model's evidence remains human-consistent.

Why is SALICON challenging?: SALICON's "human saliency" is derived from mouse tracking rather than eye fixations, yielding coarser, noisier, and more center-biased maps with higher inter-subject variance Tavakoli et al. (2017). Combined with COCO's cluttered, multi-object scenes and ImageNet-trained backbones that emphasize a single discriminative object, clean Grad-CAM often only partially overlaps the "human" map. This compresses the clean–vs–patched alignment margin and weakens alignment as a detector signal.

Why can WA+Grad outperform WA+Grad+Align?: When the saliency reference is noisy or mismatched (mouse proxies or a saliency model under domain shift), the alignment term introduces variance and false penalties. Meanwhile, weighted activations and gradient statistics already capture patch-induced brittleness. Given Grad-CAM's coarse spatial resolution and layer sensitivity, the alignment cue can be destabilizing; in such regimes, the two-cue detector is more reliable.

## 4.2 QUALITATIVE ANALYSIS

Figure 1 illustrates examples of our detections. For clarity, we present visualizations in a vertical pairing format, where each benign image and its associated interpretability maps (Grad-CAM and human saliency prediction maps, both binarized using a 75-percentile thresholding) are followed by their adversarially patched counterpart. This row-wise arrangement facilitates a step-by-step comparison of the model's interpretive shift under attack. In a benign example, the Grad-CAM highlights the true object and overlaps substantially with the human saliency. In contrast, in the patched image, Grad-CAM concentrates on the adversarial patch that has minimal human saliency. Our detector correctly flags the latter as attacked.

## 4.3 FEATURE ABLATION STUDY

We conducted ablation experiments to quantify the importance of each feature group. In Table 1 (and additional ablation figures), we see that each category of features contributes significantly to the detector's success: leaving out the gradient-based features or leaving out the saliency-divergence feature causes a noticeable drop in performance. For instance, excluding the human-saliency alignment cue reduces F1 from 99.64% to 84.01% on CAT2000/ResNet-50 (WA+Grad) and from 99.09% to 91.90% on CAT2000/EfficientNet-B0 (WA+Grad). It also lowers the true positive rate at low false-positive levels. The layer-sensitivity features are somewhat correlated with the gradient magnitude features (since both reflect the model's response strength), but they still add complementary signal. Interestingly, even without using the human saliency map, patched images still stood out, indicating that the patch draws the model's attention in ways that deviate from normal image characteristics. Nevertheless, using the true human saliency gave a much sharper separation between clean and patched images, validating our use of eye-tracking data in the detection process.

## 4.4 ROBUSTNESS TO ADAPTIVE ATTACKS

Robustness to saliency-aligned adaptive patches. We implemented a saliency-aligned adaptive objective that augments the targeted patch loss with an alignment term encouraging the model's

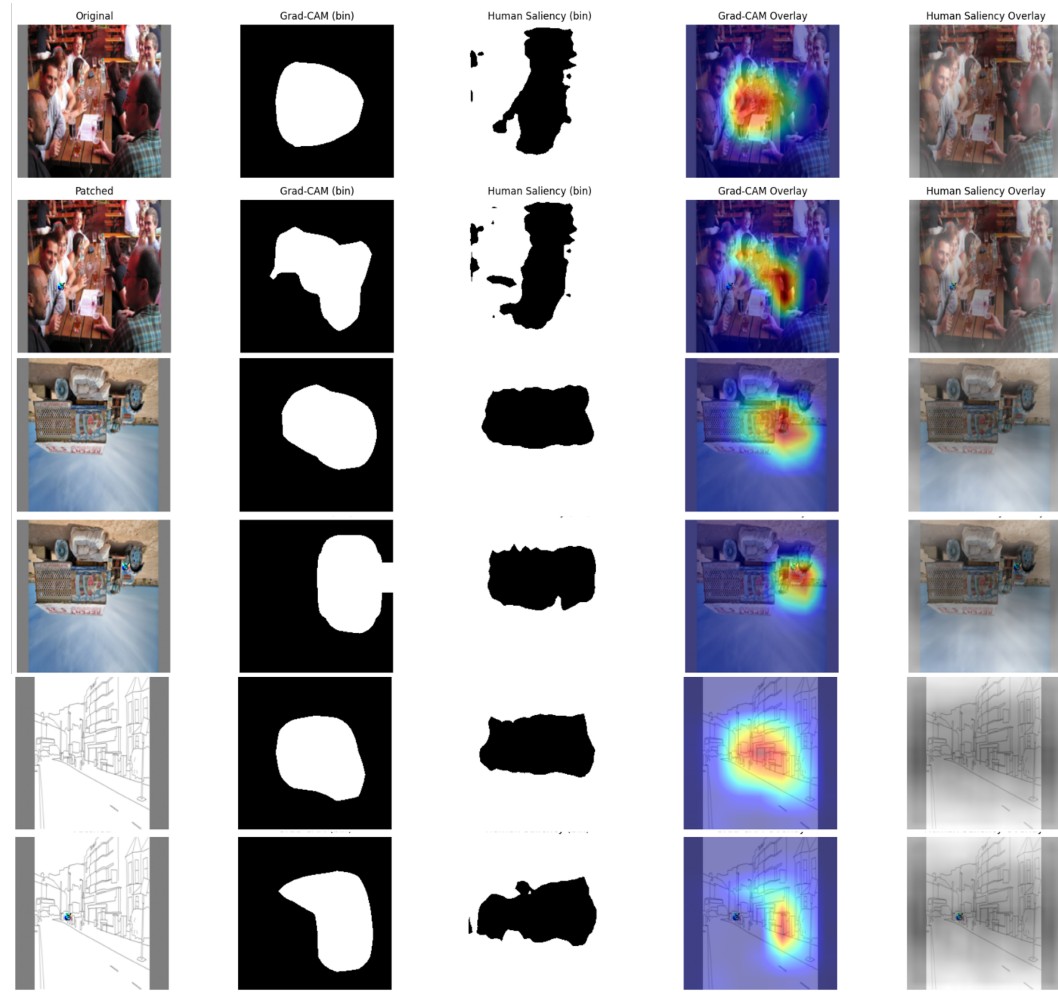

Figure 1: Comparison of binarized Grad-CAM and human saliency overlaid on original images vs. patched images from CAT2000 dataset. Grad-CAM focuses on high-level features, while human saliency captures attention-grounded global regions.

Grad-CAM to correlate with a human saliency predictor (DeepLabV3-ResNet50 retrained for FI-GRIM). Concretely, the optimizer minimizes cross-entropy toward a fixed target class while rewarding higher Grad-CAM–to–human agreement on the patched image. We evaluated this on FIGRIM with EfficientNet-B0 (21 classes; 508 train / 128 test images after our split). The resulting universal adaptive patch achieved a targeted success rate of 0.50 on the test split—i.e., only half of patched test images were forced into the target label—indicating a substantial reduction in attack reliability under alignment pressure.

On the detection side, we trained the proposed detector on the same FIGRIM setting (weighted activations + gradient statistics + saliency alignment features). Averaged over per-class detectors, we obtained precision = 0.987, recall = 1.000, and F1 = 0.993 on held-out data, showing that the detector remains highly effective even when the attacker explicitly optimizes to nudge the model's explanation toward human-like focus. Empirically, pressing the patch to appear more "human-aligned" did not remove the tell-tale footprints our detector exploits: either the patch's influence diffuses (weakening the targeted effect, as reflected in the 0.50 success rate), or its gradients/activation sensitivities remain atypical enough to be flagged.

Taken together, these results support our central claim: attempts to cloak a patch by aligning model saliency with human attention introduce a trade-off that either degrades targeted efficacy or fails to normalize internal signals picked up by our features. While a broader sweep of alignment weights and architectures is possible, the FIGRIM–EfficientNet-B0 study already demonstrates strong detector robustness against this class of adaptive strategies.

### 4.5 Transferability and Generalization

Our current detector is trained and evaluated on a specific base classifier (ResNet-50). In practice, one might want a detector that generalizes across different model architectures. Since our features are model-dependent (gradients, and activations), a detector trained on one architecture might not directly apply to another. However, the concept of what indicates a patch attack should transfer. In future work, one could retrain the detector on features from a new model, or even train a single detector on a mixture of models' data to create a more model-agnostic tool. In our project, we performed a preliminary exploration by training the detector on adversarial patches generated for one model and testing it on another model's patched images. We found that many feature anomalies persist across models. For example, a patch that breaks ResNet-50 also tends to cause a large Grad-CAM divergence and high gradient norms on other CNN-based models due to adversarial transferability Tramèr et al. (2017), but the detection thresholds might need adjustment per architecture. This suggests that while our detector can generalize conceptually, some calibration is required for different models.

Our results demonstrate that the proposed method is competitive or superior across datasets and backbones. Alignment consistently helps on FIGRIM and EfficientNet-B0, while on SALICON (mouse-proxy saliency) WA or WA+Grad can edge out the full model. Overall, we reach up to 99.64% F1 on CAT2000 (ResNet-50), 94.23% F1 on FIGRIM (EfficientNet-B0), and 96.01% F1 on SALICON (ResNet-50), while providing human–model alignment diagnostics that baselines lack. This reinforces our core claim: model interpretability, when correctly harnessed, enhances, rather than undermines, adversarial robustness. Our method also offers richer qualitative interpretability than prior work, as seen in LIME analyses and Grad-CAM visualizations where detection decisions are grounded in recognizable visual patterns, not black-box scores.

## 5 Limitations and Future Scope

One potential concern is the reliance on human saliency data. In practice, obtaining eye-tracking maps for every image is not feasible. We address this by noting that alternative sources of "expected attention" can be used (for instance, saliency prediction models or simple priors), and by referencing related work in robust ML where incorporating human attention priors improved model reliability. Finally, while our detector in its current form is tuned for visible patch attacks, the framework is flexible. Features can be added or adjusted to target other attack types (for example, distributed adversarial noise or camouflage attacks). In future work, once an attack is detected, one could also explore downstream defenses. For instance, refusing to classify the image, alerting a human operator, or attempting to remove the patch before feeding the image to the classifier. We can further employ interpretability techniques (such as LIME Ribeiro et al. (2016)) to explain the detections and enhance user trust. Interpretability evaluation techniques like ROAR Hooker et al. (2019) and works such as Wang et al. (2022) emphasize the utility of removing important regions and retraining to assess attribution quality. Our saliency-based ablations follow a similar rationale by testing performance drops when key interpretability signals are withheld. This can be further explored to improve the robustness rather than mere detection.

## 6 Conclusion

Adversarial patch attacks threaten visual classifiers by causing misclassification via a small, crafted image region, often without altering human perception. We propose a probe-based detection framework that fuses gradient statistics, layer-wise activation sensitivity, and explicit saliency alignment between model and human attention. Lightweight probes attach to a pretrained, frozen classifier to extract gradient moments (mean, max, min, std), each layer's perturbation sensitivity, and divergence between model focus (Grad-CAM) and human saliency. A downstream classifier is trained on these features to label inputs as benign or patched. The approach is model-agnostic and leaves the base classifier unchanged. On CAT2000, FIGRIM, and SALICON, our detector achieves state-of-the-art detection with minimal false alarms. Overall, combining gradient, sensitivity, and saliency-alignment cues yields a robust, explainable defense against localized patch attacks.

## 7 REPRODUCIBILITY STATEMENT

We provide a anonymized code artifacts in the supplementary material to reproduce the results: `https://anonymous.4open.science/r/patch_sage-D444`.

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
