# Supplementary Material

## A  Appendix: Implementation Details

### A.1  Adversarial Patch Creation Process

Patch Attack Settings: We generate adversarial patches following the threat model of Brown et al. (2017), where a visible square patch is inserted into an input image to induce misclassification. In our implementation, the patch occupies approximately 20% of the input image area by default (e.g., 45×45 pixels on a 224×224 image). The patch is initialized with random values and is applied at random locations within each image in a batch. For each image, we uniformly sample a top-left coordinate that ensures the patch remains fully within frame. This placement strategy avoids trivial occlusions of the main object, allowing humans to still recognize the object, while the model is misled by the adversarial signal.

Attack Objective: We primarily generate targeted adversarial patches. The goal is to force the classifier to consistently predict a chosen target class, regardless of the true label of the input. For a model $f$ with output logits $f(x)$, the loss function for targeted attacks is defined as the cross-entropy between $f(x)$ and a fixed target label $t$: $L_{target}(x) = CE(f(x), t)$.

Optimization Method: We adopt a gradient-based optimization procedure, directly updating the patch pixels as learnable parameters. The patch tensor $P \in \mathbb{R}^{3 \times h \times w}$ is initialized randomly and optimized using the Adam optimizer with a learning rate of $5 \times 10^{-2}$. At each training step:

- A batch of images $x_i$ is sampled from the dataset.
- The patch is applied at a random location within each image using a placement function, producing $x_i^{\text{patched}}$.
- The patched images are forwarded through the classifier to obtain logits $f(x_i^{\text{patched}})$.
- The loss $L$ is computed according to the attack mode i.e. targeted.
- Gradients are backpropagated only to the patch tensor $P$, and the Adam optimizer updates the patch values.

This process is repeated for 300 iterations, and the patch parameters are clipped to valid image ranges after each step. Unlike $L_p$-bounded perturbations, our patch is unconstrained in pixel norm but restricted to its fixed spatial mask.

### A.2  Experimental Design and Implementation Details

#### A.2.1  Base Classifier Models

For all experiments, we selected two widely used convolutional neural network architectures as the victim classifiers: ResNet-50 and EfficientNet-B0. Both models were pretrained on ImageNet and fine-tuned on the datasets. ResNet-50 is a deep residual network with skip connections, offering robust feature hierarchies, while EfficientNet-B0 is a lightweight architecture designed via neural architecture search with compound scaling, providing strong accuracy-efficiency trade-offs. Using both allowed us to test whether our detector generalizes across architectures with different design philosophies.

#### A.2.2  Datasets and Human Saliency Ground-Truth

We evaluated our detector on three benchmark datasets that provide human visual saliency data:

- CAT2000 Borji & Itti (2015): Contains 2,000 natural images spanning 20 categories. Ground-truth saliency maps are derived from eye-tracking experiments, making this a reliable measure of human visual attention. Because CAT2000 includes categorical groupings, we leveraged these to contextualize model predictions.

- FIGRIM Bylinskii et al. (2015): Contains images collected for memory and gaze studies, with ground-truth saliency maps also obtained from eye-tracking.

- SALICON Jiang et al. (2015): A large-scale dataset with 10,000 training and 5,000 validation images, originally derived from MS-COCO. Instead of eye-tracking, it uses mouse-tracking approximations of human fixations, which, while less precise, correlate strongly with real gaze behavior. Unlike CAT2000, SALICON does not include object category labels. To enable classifier-based evaluation, we assigned categories by running a pretrained ImageNet classifier on each image and selecting the top-1 predicted label if its confidence exceeded a threshold. This ensured consistency across our pipeline, though we acknowledge this introduces some label noise.

Together, CAT2000 and FIGRIM provide high-validity gaze-based ground truth, while SALICON offers scale and category labels with slightly noisier attention maps.

### A.2.3 SALIENCY MODEL TRAINING

Since experiments required predicted saliency maps, we trained human saliency prediction models on each dataset. For training and validation, we followed an 80/20 split. We ensured that the resulting saliency models achieved an AUC score greater than 0.75 on their respective test splits, validating that predicted maps aligned sufficiently with human fixations. This quality check was necessary to ensure downstream reliability in saliency alignment-based detection.

### A.2.4 CLASSIFIER TRAINING

To enable classifier-based evaluation, we did two things:

- Category label assignment: Because SALICON lacks object class labels, we used a pretrained ImageNet classifier to assign a pseudo-label to each image. Specifically, we took the classifier's top-1 predicted class for each image, conditioned on the prediction confidence exceeding a threshold. These pseudo-labels allowed us to align SALICON images with ImageNet categories and evaluate patch attacks in a comparable manner to CAT2000 and FIGRIM.

- Train/test split: We created a stratified 80/20 split of the FIGRIM dataset across its scene categories, ensuring that both training and test subsets preserved the dataset's distribution.

This setup ensured that all three datasets were usable in a unified experimental framework, while acknowledging FIGRIM's limitations (pseudo-labels rather than human-provided categories).

### A.2.5 DETECTOR TRAINING PROTOCOL

For each dataset, we constructed paired datasets of benign and patched images. For each dataset, we sampled a pool of images and generated adversarial patch variants. In each case, we maintained a 50/50 class balance (clean vs attacked) to avoid bias.

### A.3 PROBE MODULES AND FEATURE EXTRACTION

### A.3.1 NON-INTRUSIVE PROBES

We attach lightweight probes to the base classifier using forward hooks on its convolutional blocks. These probes record intermediate activations and gradients but do not alter the forward inference of the model. Thus, the classifier remains frozen and unaffected, while the probes provide analytical signals for downstream detection.

### A.3.2 GRADIENT STATISTICS

For each input image, we compute the gradient of the model's predicted class logit with respect to the input pixels. Specifically, let $y_{\text{pred}} = \arg\max_c f_c(x_{\text{clean}})$ be the model's top-1 prediction on the clean image. We then calculate $g = \nabla_x . f_{y_{pred}}(x)$ both for the clean and patched versions of the image. This ensures that gradients are always referenced to the classifier's original (intended) prediction. From $g$, we extract four statistics — mean, standard deviation, maximum, and minimum gradient values — as scalar features. These statistics capture how gradients become sharper and more localized under adversarial patches, in contrast to the smoother, more distributed gradients of clean images.

### A.3.3 LAYER-WISE SENSITIVITY

We measure the robustness of the classifier to perturbations at different layers using noise injection hooks. For each convolutional block $\ell$, we store the clean activation norms, then inject Gaussian noise into that layer's activations and compute the effect on the output confidence distribution: $\nabla_\ell = \|softmax(f(x)) - softmax(f(x_{pert}(\ell)))\|_2$

where $x_{\text{pert}}^{(\ell)}$ denotes the forward pass with perturbed activations at layer $\ell$. The resulting sensitivity vector $(\Delta_1, \Delta_2, \ldots, \Delta_L)$ is scaled by the original activation norms, providing a set of sensitivity features. Intuitively, adversarial patches induce spiky sensitivities at specific layers (often high-level layers dominated by patch patterns), whereas clean images yield more stable, diffuse sensitivities.

Saliency Alignment (PatchSAGE): We compute the alignment between model saliency and human saliency maps.

- For the model, we use Grad-CAM on the final convolutional layer with respect to $y_{\text{pred}}$.
- For the human reference, we use ground-truth or predicted saliency maps (e.g., from trained saliency models).

Both maps are normalized to probability distributions. We then compute:

- Pearson correlation ($\rho$): Measures linear correspondence between maps.
- Intersection-over-Union (IoU): Obtained by thresholding the top 20% salient pixels in each map and computing overlap.

To make the alignment discriminative, we calculate the IoU difference $1 - $ IoU as a feature (higher when model focus diverges from human focus). In patched images, Grad-CAM highlights the adversarial patch while human saliency remains on the true object, resulting in near-zero IoU and weak correlation. Clean images show moderate positive correlation and partial overlap.

Feature Fusion and Detector Training: We construct a feature vector for each image by concatenating:

- Layer sensitivities (per-block deltas × activation norms)
- Gradient statistics (4 scalars)
- Saliency alignment (IoU difference)

These are combined using a weighted scheme (e.g., $0.85\times$ sensitivity features + $0.1\times$ gradient features + $0.05\times$ IoU difference). The resulting feature vectors are used to train per-class binary Random Forest classifiers (10 estimators, scikit-learn default hyperparameters). Each classifier distinguishes clean from patched samples within its class. Before training, features are standardized using z-score normalization.

We evaluate detection using precision, recall, F1-score, and AUC, reporting both per-class and average performance across categories. To ensure reproducibility, we save trained classifiers along with their feature scalers as serialized .pkl objects.

### A.4 REPRODUCIBILITY ADDENDUM

Environment:

Python 3.10; PyTorch 2.x; torchvision 0.15+; scikit-learn 1.3+; OpenCV-python 4.x; pytorch-grad-cam 1.4+. Experiments run on a single NVIDIA GPU (e.g., V100/A100/RTX). We fix torch, numpy, and dataloader seeds to 42 and enable deterministic flags where applicable.

Datasets & splits: CAT2000 (eye tracking): we use the official train/test split (no cross-split mixing). FIGRIM (eye tracking): lacks class labels and official splits. We pseudo-label each image with the top-1 prediction of a pretrained ImageNet classifier (confidence ¿= 0.5) and perform a stratified 80/20 split by pseudo-class. SALICON (mouse tracking): class labels are taken from the corresponding MS-COCO annotations (image–category mapping). We use the official train/val split.

All images are resized to 224×224 and normalized with ImageNet mean/std. We release the exact file lists (train/val/test) used in our runs.

Base classifiers: Backbones: ResNet-50 and EfficientNet-B0 (ImageNet-pretrained). For each dataset we fine-tune the last classifier head (and, when noted in configs, the last conv stage) with SGD (momentum 0.9, weight decay 1e-4) or Adam; batch size 64; 30 epochs; cosine LR schedule (base LR 1e-3 for the head, 1e-4 when unfreezing one conv stage). We provide the exact config used per (dataset, backbone).

Human saliency (ground truth & prediction): Ground truth: CAT2000/FIGRIM use eye-tracking maps; SALICON uses mouse-tracking maps. Predicted saliency (for alignment features): DeepLabV3-ResNet50 with a 1-channel head (BCEWithLogits). Train/val split 80/20 within each dataset; input 224×224; Adam (LR 1e-4), batch 16, 20 epochs, light aug (flip, color jitter). We require AUC ¿= 0.75 on the held-out split before using a checkpoint for detection features. Checkpoints are released.

Adversarial patches: Form: visible square patch applied at a random valid location per image. Size: side S = floor(0.20 × 224) = 45 px. Objective: targeted CE toward a fixed ImageNet class; universal patch optimized over batches. Optimizer: Adam, LR 5e-2, 300 steps; values clipped to normalized bounds after each step. We ensure no image used to optimize a universal patch appears in the detector test set.

Feature extraction (probes): The code can be found here: `https://anonymous.4open.science/r/patch_sage-D444/train_figrim_res50.ipynb` Gradients: for the clean image's top-1 class, compute input-gradient map and store mean, std, max, min. Layer sensitivity: register forward hooks on conv blocks; for each block, add Gaussian noise ($\sigma$=0.1) to the activation and record the difference. Multiply the difference by the clean activation norm of that block. Saliency alignment (PatchSAGE): Grad-CAM on the final conv layer (predicted class) vs. human saliency; both normalized to probability maps. Features: IoU of top-20% pixels (we use 1-IoU).

Detector & evaluation For each (dataset, backbone) and (optionally) per pseudo-class, we z-score features and train a RandomForest (number of estimators is 10, random state is 42). Class balance is 1:1 (clean vs. patched). We report precision/recall/F1/AUC on held-out data (default detector split 70/30, random state=42). We release scalers, trained RFs, and evaluation logs.

Baselines We provide code implementations of ProbeNet, SentiNet, and X-Detect (Grad-CAM layer, segmentation/occlusion parameters) to make our Table-1 rows exactly reproducible: `https://anonymous.4open.science/r/patch_sage-D444/related_figrim_eff.ipynb`

### A.5 RELEVANT WORKS - NOT COMPARED AGAINST

On Detecting Adversarial Perturbations Metzen et al. (2017) introduced an early detection approach wherein a small "detector" subnetwork is attached to a hidden layer of the classifier to distinguish normal inputs from adversarially perturbed ones. We did not include a separate comparison to this method because our evaluation already contains ProbeNet Rounds et al. (2020), a baseline that is a strict extension of Metzen's idea. ProbeNet generalizes the single-layer detector by probing multiple internal features, effectively subsuming the original approach. Including Metzen's detector would thus be redundant, as ProbeNet's results represent the stronger version of that strategy.

Minority Reports Defense: The Minority Reports method McCoyd et al. (2020) detects patch attacks by systematically occluding different regions of the image and observing the classifier's predictions. If some occlusion placements fully cover an adversarial patch (revealed by a cluster of "correct" predictions that disagree with the majority), the system flags the input as attacked. While conceptually

powerful – it can even provide certified robustness for bounded patch sizes – we found it impractical to deploy in our setting. Achieving high detection fidelity with Minority Reports requires scanning a large patch search space at fine stride, leading to enormous computational overhead. In fact, the original authors report an inference cost roughly 900× higher than a normal forward pass on CIFAR-10 when using a full occlusion grid. This makes real-time or large-scale evaluation infeasible, so we exclude Minority Reports from direct comparison.

### A.5.1 EXPLANATION-DRIVEN DETECTORS

Several prior works approach adversarial detection through shifts in model explanations. Schut & Gal introduce a learned saliency model trained separately from the classifier, then detect attacks by checking whether the explainer's attribution disagrees with the classifier's evidence. On MNIST and CIFAR-10 this method surpasses raw gradient saliency as a detector. However, it requires training an additional explainer and is designed mainly for small-norm perturbations rather than visible patches, making it mismatched with our test-time, retraining-free, patch-focused setting.

Wang & Gong (2022) (ML-SAFE) extract multi-layer saliency maps (e.g., Grad-CAM, guided back-propagation at multiple depths) and aggregate them into features for a detector. They show that adversarial and clean inputs diverge in deeper attributions. While effective for general adversarial examples, ML-SAFE requires computing numerous explanation maps per image, introducing significant overhead. By contrast, our approach fuses lightweight gradient/sensitivity cues with human–model alignment and directly targets patch attacks. Given this difference in threat model and computational cost, we do not treat ML-SAFE as a direct baseline.

Sim & Song (2025)) propose combining Grad-CAM with clustering metrics (e.g., silhouette scores) to detect unusually concentrated attention as adversarial. Their study centers on CIFAR-10 classifiers and ensembles, and detection relies on unsupervised clustering over attribution maps. Our setting diverges in several ways: ImageNet-scale backbones, human–model saliency alignment, and patch-focused threats. Moreover, reproducing their clustering/ensemble pipeline would add substantial runtime overhead without aligning with our emphasis on single-pass features for large-scale evaluation. We consider this line complementary rather than a direct comparison.

In summary, these explanation-based detectors are conceptually aligned with our use of saliency but generally (i) focus on small-norm perturbations instead of visible patches, (ii) require extra models or multiple saliency maps per input, or (iii) operate in small-dataset regimes. Our work is distinguished by its emphasis on patch attacks, no retraining, human–model alignment, and low per-image overhead, which is why we do not position these methods as direct competitors.

### A.6 ABLATION STUDIES AND FUTURE WORK

Completed Ablations We carried out ablation experiments focusing on the role of sensitivity and gradient features, as well as their combination with saliency alignment:

- Sensitivity Only: Using only the layer perturbation sensitivity vector, we observed moderate detection performance. Patched images often produced anomalous spikes in sensitivity at higher layers, but certain clean images near decision boundaries were sometimes misclassified.

- Sensitivity + Gradient: Combining sensitivity with gradient statistics yielded stronger performance than either alone. On CAT2000, this setup reached F1 scores around 90%, confirming that the two cues are complementary. However, without saliency alignment, some subtle or adaptive patches remained undetected.

- Sensitivity + Gradient + Saliency (Full Model): The complete feature set provided the best results, with F1 scores near 98–99% on both CAT2000 and SALICON. Each feature type addressed different aspects of adversarial behavior, and their fusion proved essential for robustness. Excluding any one feature led to noticeable drops in both overall F1 and performance at low false-positive rates, underscoring the critical role of saliency alignment in particular.

Future Ablations Several additional ablation configurations remain to be explored, which we identify as promising future work:

- Gradients Only: While patched images often produce sharp gradient spikes, evaluating this cue in isolation could quantify how much detection signal lies purely in first-order sensitivity.
- Saliency Alignment Only: Since human–model saliency misalignment is highly discriminative, testing it as a standalone signal would provide insights into its limits (e.g., in cases where human and model focus diverge naturally).
- Gradient + Saliency: This combination may capture both internal perturbation footprints and external focus misalignment, potentially rivaling the full model in power.
- Sensitivity + Saliency: Pairing internal robustness cues with saliency alignment could provide a strong alternative to gradient-based signals, especially against adaptive attacks.

We leave these ablations for future extensions, as they would further clarify the complementary roles of each feature type.

## A.7 LIMITATIONS AND FUTURE WORK

### A.7.1 DEPENDENCE ON HUMAN SALIENCY DATA

Our approach relies on human saliency maps to measure alignment with model explanations. While this signal proved powerful, it requires either ground-truth gaze data or high-quality saliency predictions. Collecting eye-tracking data at scale is impractical, and mouse-based approximations such as SALICON are noisier. We mitigated this by training saliency prediction models on each dataset, ensuring AUC ¿ 0.75, but prediction errors remain a source of variability. Future work should explore incorporating stronger saliency models into the detection pipeline, evaluating trade-offs between accuracy and cost, and investigating which aspects of human attention (e.g., foveal vs. peripheral) are most critical. Domain-specific or user-specific gaze priors may also help tailor detectors to specialized applications.

### A.7.2 ABLATION COVERAGE

Our ablation analysis focused on sensitivity-only, sensitivity + gradient, and the full model (sensitivity + gradient + saliency). These confirmed that each feature type contributes unique information and that their fusion yields the strongest results. However, we did not test gradients-only, saliency-only, or the remaining pairwise combinations (e.g., gradient + saliency, sensitivity + saliency). Future work should explore these systematically to better quantify the complementary roles of each feature type, especially in adaptive attack settings.

### A.7.3 GENERALITY TO OTHER ATTACKS

The detector is designed for visible, localized adversarial patches where model attention shifts away from human focus. Other attack types—such as blended camouflage patches, distributed noise, or temporal attacks in video—may not trigger the same alignment divergences. While gradient and sensitivity features may still capture anomalies, our framework has not yet been extended to such cases. Future work should adapt the approach to cover a broader range of threats, potentially by adding global perturbation measures, temporal consistency checks, or multi-modal alignment cues.

### A.7.4 TRANSFERABILITY ACROSS MODELS

While we evaluated our detector on two backbone architectures (ResNet-50 and EfficientNet-B0), we did not explicitly study transferability across architectures. In practice, an adversary may attack a surrogate model and deploy the patch against another unseen target model. Future work should therefore assess whether detectors trained on one architecture generalize to different ones. This experiment would clarify the robustness of probe-based detection under model mismatch and help in designing universal detectors.

### A.7.5 PHYSICAL-WORLD EVALUATION

Our experiments were conducted in the digital setting. However, adversarial patches can also be printed and placed into physical scenes. These introduce new challenges, including lighting vari-

ation, viewpoint changes, and camera noise. Future work should evaluate the detector in such physical-world scenarios, testing whether alignment features remain discriminative when patches are subject to real-world transformations. This would strengthen the case for deployment in safety-critical domains like surveillance or autonomous driving.

### A.7.6 POST-DETECTION MITIGATION

Currently, the detector outputs a binary decision (clean vs. patched) without suggesting corrective actions. In practice, mitigation may be desirable: e.g., removing or inpainting the suspected patch region before reclassification. Although we do not explicitly localize patches, Grad-CAM often highlights the patch region when an attack is detected. Leveraging this for patch removal or restoration represents a promising detect-and-mitigate pipeline. Future experiments should quantify how well such corrections restore classifier accuracy without harming clean images.

### A.7.7 INTERPRETABILITY AND TRUST

An appealing feature of our approach is interpretability: the detector relies on saliency maps, gradient signals, and sensitivity measures that can be visualized and explained. Future work could deepen this aspect by applying tools like LIME, SHAP, or ROAR to the detector's features, showing which cues drove its decision. This would help build user trust (e.g., highlighting that an image was flagged because model focus diverged from human focus in an unusual region).

### A.7.8 EFFICIENCY AND DEPLOYMENT

Our pipeline requires a backward pass to compute input gradients, plus a small number of additional forward passes for layer perturbations; the overall cost scales with the number of probed layers. Future work should profile runtime costs and explore optimizations such as distilling the detector into a lightweight neural model, reducing redundant probe features, or applying the detector selectively (e.g., only when quick heuristics flag an input as suspicious). Balancing robustness with efficiency will be essential for deployment in latency-sensitive domains like autonomous driving.

### A.7.9 BROADER APPLICABILITY

Finally, the principle of human–model alignment extends beyond adversarial patch detection. We anticipate its relevance in settings such as detecting data poisoning or backdoor triggers, where misalignment between human and model focus may reveal hidden manipulations. Extending this framework to multi-modal models (e.g., image–text systems) may also help identify incoherent or adversarially crafted inputs.