# OpenReview forum: "PatchSAGE: A Probe-Based Detector Using Saliency Alignment, Gradients, and Layer Sensitivity"
_ICLR.cc/2026/Conference — ICLR 2026 Conference Withdrawn Submission_

### Official Review · Reviewer_FtTp · 2025-10-26

**Soundness:** 2
**Presentation:** 2
**Contribution:** 2
**Rating:** 2
**Confidence:** 4

**Summary:**

This paper introduces PatchSAGE, a post-hoc detector for adversarial patch attacks. It attaches lightweight probes to a frozen vision model to extract gradient statistics, activation sensitivity, and saliency alignment features, which are fed into a small classifier to detect patched images. Experiments on CAT2000, FIGRIM, and SALICON with ResNet-50 and EfficientNet-B0 show F1 scores of 82–99%, outperforming SentiNet and X-Detect.

**Strengths:**

1. Clear, modular setup: post-hoc, no retraining, easy to reproduce.

2. Interpretable components: gradients and Grad-CAM visualization offer qualitative insight.

3. Reasonably strong empirical F1 on simple datasets (CAT2000, FIGRIM).

**Weaknesses:**

1. Limited novelty: essentially re-uses gradient magnitude, activation perturbation, and Grad-CAM misalignment as features.

2. Dependence on human saliency data: impractical for deployment and limits reproducibility outside saliency datasets.

3. Overstated results: comparisons are against weak baselines; stronger detectors (e.g., certified or ensemble-based methods) are not tested.

4. Weak evaluation of adaptivity: the “saliency-aligned adaptive patch” experiment is narrow and attacker-dependent, not a robust stress test.

5. No generalization analysis: the detector is retrained per model; cross-architecture robustness is minimal.

**Questions:**

1. How does PatchSAGE perform under adaptive attacks that directly optimize the gradient-, sensitivity-, and alignment-based features jointly?

2. Since human saliency maps are unavailable in most domains, can the method remain effective using only model-based saliency surrogates?

---

### Official Review · Reviewer_mBbP · 2025-10-30

**Soundness:** 2
**Presentation:** 2
**Contribution:** 2
**Rating:** 2
**Confidence:** 5

**Summary:**

- The paper proposes a detection pipeline for adversarial patches built from three probe families on a frozen classifier: gradient-based summary stats, layer-wise activation sensitivity, and human–model saliency alignment (Grad-CAM vs. human saliency). Reported F1 is high on CAT2000, FIGRIM, and SALICON.

**Strengths:**

- Adversarial patch attacks remain one of the most practical and concerning threats to deep vision systems. The paper’s focus on detecting such attacks contributes to an important and underexplored direction, especially toward practical, interpretable defenses rather than purely algorithmic robustness.

**Weaknesses:**

- The method assumes that the detector knows the clean image’s original prediction when computing the gradient-based features. In practice, this information isn’t available during deployment, only the potentially attacked (patched) image is. This means the detector relies on knowledge it wouldn’t realistically have in a real-world setting. The authors should either demonstrate that their approach works without this assumption or provide a clear justification for why access to the clean label is reasonable.
- The evaluation focuses on probe-based detectors (e.g., SentiNet, ProbeNet) but omits comparisons with recent patch defense frameworks (e.g., PatchGuard, Diffender, or ODDR) or adversarially trained models. Including these would provide a stronger sense of how PatchSAGE performs relative to current state-of-the-art defenses.
- A stronger evaluation would include white-box attacks that directly optimize against the detector’s features (e.g., differentiating through Grad-CAM, sensitivity probes, or surrogate models), attacks tested under more aggressive transformation settings, and variants that disguise gradient statistics or mimic human saliency through alternative priors. Finally, extending the analysis to physical-world adaptive patches would make the claims about robustness more convincing.
- Some of the datasets used rely on approximate or indirect forms of supervision, such as noisy saliency maps or classifier-generated pseudo-labels. These choices can introduce uncertainty and potential bias into the evaluation, making it harder to isolate the true effect of the proposed method. The authors should examine how sensitive their results are to these factors. For example, by testing alternative labeling strategies or using different saliency predictors to ensure the conclusions remain consistent across data sources.
- The proposed method involves additional computation (including backward passes and layer-wise perturbations) yet the paper doesn’t report any runtime, throughput, or GPU memory measurements. Since these probes could add significant overhead, especially in real-time or large-scale settings, it’s important to quantify the computational cost. The authors acknowledge this limitation, but providing concrete efficiency metrics would greatly strengthen the paper’s practical relevance.

**Minor weaknesses:**
- Several references are missing or incomplete (for example, around lines 85 and 100).
- The paper overuses em dashes, which affects readability. Replacing them with commas or parentheses where appropriate would improve clarity and presentation quality.

**Questions:**

Please check weaknesses.

---

### Official Review · Reviewer_QUAg · 2025-10-31

**Soundness:** 3
**Presentation:** 2
**Contribution:** 2
**Rating:** 6
**Confidence:** 3

**Summary:**

The paper introduces PatchSAGE, a model-agnostic framework for detecting adversarial patches by combining gradient statistics, layer sensitivity analysis, and human–model saliency alignment. ​ It achieves state-of-the-art detection performance with F1 scores up to 99.64% across datasets and models, outperforming existing methods like SentiNet and X-Detect, while offering interpretable detection rationales. ​
PatchSAGE integrates human saliency alignment with gradient statistics and layer sensitivity analysis to detect adversarial patches effectively. ​ It is model-agnostic, requires no retraining, and achieves high accuracy across diverse datasets (CAT2000, FIGRIM, SALICON) and architectures (ResNet-50, EfficientNet-B0). ​ The framework enhances transparency by providing interpretable rationales for detection decisions.

**Strengths:**

- The proposed PatchSAGE framework delivers impressive results, achieving F1 scores up to 99.64% across multiple datasets (CAT2000, FIGRIM, SALICON) and architectures (ResNet-50, EfficientNet-B0). It clearly surpasses prior methods such as SentiNet and X-Detect, setting a new benchmark for adversarial patch detection.
- The method is model-agnostic and functions as a post hoc detector, requiring no retraining or architectural changes to the base classifier. This design makes it highly practical and easy to integrate into existing vision systems.
- A notable strength lies in the use of human saliency alignment, comparing Grad-CAM attention maps with human attention data. This enables effective identification of adversarial patches that exploit discrepancies between human and model focus—a thoughtful and biologically inspired approach.
- The framework combines saliency alignment, gradient statistics, and sensitivity analysis to produce interpretable detection explanations. This focus on transparency enhances user trust and contributes to broader understanding of model behavior under attack.

**Weaknesses:**

- The approach relies heavily on human saliency maps, which may not always be available for all datasets or domains. Although saliency prediction models can serve as substitutes, they can introduce noise or inaccuracies that may degrade detection performance.
- The detector is trained on features tied to specific model architectures, potentially restricting its applicability to other models without additional retraining or calibration efforts.
- In datasets such as SALICON, where human saliency is inferred from mouse tracking rather than eye fixations, the alignment signal may be noisy and biased, diminishing the robustness of the saliency alignment component.
- The framework primarily targets visible patch-based adversarial attacks. Its effectiveness against other attack types—such as distributed perturbations or camouflage-style attacks—remains limited and would require further extension or adaptation.

**Questions:**

- The reliance on human saliency maps limits practicality, as such data are rarely available. Could the authors clarify how predicted saliency maps (e.g., DeepGaze, SAM) could be used without degrading performance? Including experiments or analysis on robustness to noise or domain shifts in these proxies would strengthen the paper’s real-world relevance.
- The paper briefly mentions cross-model transferability (e.g., ResNet-50 → EfficientNet-B0). More details on how well the detector generalizes to unseen models, and how much calibration or retraining is required, would clarify its broader applicability. A discussion on developing a model-agnostic variant would further enhance impact.
- The adaptive patch results show a 50% drop in attack success, but it is unclear whether this holds across datasets or for attacks targeting all cues (gradients, sensitivity, alignment). Additional experiments and a discussion of computational cost and potential optimizations for faster inference would improve the method’s practicality and robustness claims.

---

### Official Review · Reviewer_Y86w · 2025-11-01

**Soundness:** 1
**Presentation:** 2
**Contribution:** 2
**Rating:** 2
**Confidence:** 4

**Summary:**

This paper proposes PatchSAGE, a post-hoc, model-agnostic detector for adversarial patch attacks. The method attaches lightweight probes to a frozen classifier and measures three complementary cues: input-gradient statistics, per-layer activation sensitivity, and alignment between model and human saliency maps. A small classifier is trained on these probe features to predict whether an input contains a patch. Across three datasets (CAT2000, FIGRIM, SALICON) and two models (ResNet-50, EfficientNet-B0), PatchSAGE outperforms baselines like SentiNet and X-Detect. The approach is simple and appears effective, but it is not comprehensively validated and relies on the availability of reliable human saliency maps, focusing mainly on visible and localized patch attacks.

**Strengths:**

- The paper provides a clear, interpretable mechanism for adversarial-patch detection using human attention modeling via saliency alignment.
- In general, the paper is well written and easy to follow.

**Weaknesses:**

- The main weakness of the paper lies in its validation and scalability. The experimental evaluation is too limited to support broad conclusions, and the method's effectiveness beyond small datasets and CNN backbones remains unclear, limiting the evidence for its generality and real-world applicability. Including experiments on larger or more diverse datasets, as well as additional architectures, would better demonstrate generalization and robustness.
- Although the approach is simple and appears effective, it depends on the availability of reliable human saliency maps and is mainly validated on visible, localized patch attacks.
- The method is described mostly at a conceptual level, without a clear formal or mathematical definition of how the different cues (saliency alignment, gradient statistics, and layer sensitivity) are integrated. A more formal presentation would improve clarity and reproducibility.
- The ablation study in Table 1 only shows cumulative results (e.g., WA → WA+Grad → WA+Grad+Align), but it does not isolate the individual contribution of each component. It would be helpful to analyze each module separately and also test intermediate combinations (e.g., WA+Align or Grad+Align) to better understand their individual and joint effects.
- More qualitative results are needed to illustrate how the method behaves in different scenarios. Only one qualitative example (Fig. 1) is provided; it would be useful to include additional visualizations and highlight failure cases to better understand the method’s limitations.

Minor comments:
- There are some missing references in the related work section (page 2).

**Questions:**

- Could the authors provide additional experiments on larger or more diverse datasets? For instance, could the method be evaluated on [Voila-A](https://github.com/naykun/Voila-A)?
- Could the authors evaluate or discuss how PATCHSAGE might perform on gaze-based datasets with real eye-tracking data, such as Ego4D, which provides task-driven human attention in dynamic scenes? This could help assess whether the proposed saliency alignment generalizes beyond static-image settings.
- Could the authors test the method on other architectures, such as transformer-based backbones (e.g., ViTs or Swin Transformers), to verify generalization beyond CNNs? If not directly, what challenges would arise when adapting the probe-based saliency alignment to non-convolutional models?
- Could PATCHSAGE detect less visible or hidden adversarial patches?
- Could the authors include ablations isolating each component individually (WA, Grad, Align) and additional combinations such as WA+Align or Grad+Align to better understand their contributions?
- Could the authors provide more qualitative examples illustrating how PATCHSAGE behaves under different attack scenarios and highlight failure cases?

---

### Note · Authors · 2025-11-18

I have read and agree with the venue's withdrawal policy on behalf of myself and my co-authors.